# Echocardiography, Computed Tomography and Magnetic Resonance Imaging in the Differential Diagnosis of a Tumor in the Left Atrium of the Heart

**DOI:** 10.3390/diagnostics12071749

**Published:** 2022-07-20

**Authors:** Malgorzata Zalewska-Adamiec, Hanna Bachorzewska-Gajewska, Slawomir Dobrzycki

**Affiliations:** 1Department of Invasive Cardiology, Medical University of Bialystok, 15-089 Bialystok, Poland; hgajewska@op.pl (H.B.-G.); kki@umb.edu.pl (S.D.); 2Department of Clinical Medicine, Medical University of Bialystok, 15-089 Bialystok, Poland,

**Keywords:** thrombus, myxoma, echocardiography, computed tomography, magnetic resonance imaging

## Abstract

Cardiac tumors are rare. Most often they are metastatic tumors, while primary tumors are much less common. In addition to proliferative changes in the heart, there are also non-neoplastic structures, such as thrombus, vegetation or inflammatory tumors. All structures with a heart tumor morphology require a lot of imaging studies in order to diagnose them and plan treatment without performing a biopsy. We present a case of a 75-year-old female patient who had moving masses in the left atrium on echocardiography. Computed tomography of the chest was performed, which did not clearly explain the nature of the structure observed in the left atrium. The Heart Team decided to perform another test—magnetic resonance imaging (MRI) of the heart in 3 months to differentiate the lesion. The examination was performed after 3 months of warfarin therapy and there were no masses in the left atrium, which confirmed that the observed tumor was a thrombus.

**Figure 1 diagnostics-12-01749-f001:**
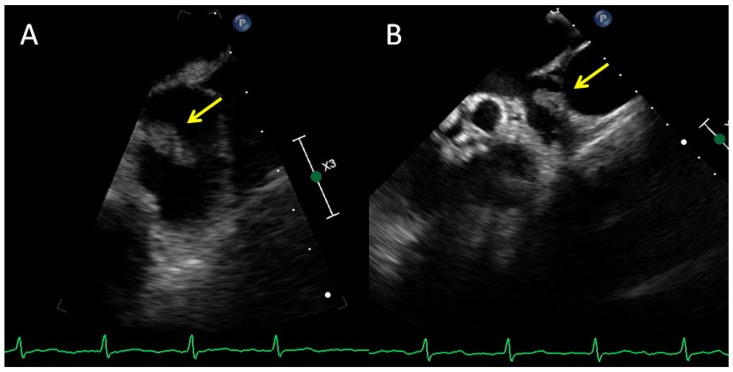
A 75-year-old female patient with permanent atrial fibrillation undergoing apixaban therapy, with a suspicion of a tumor in the left atrium, was admitted to the Clinic for further diagnosis. Transthoracic echocardiography showed good valve function and mildly impaired global left ventricular systolic function with an ejection fraction estimated at 50%. In a transesophageal echocardiographic examination in the left atrial appendage, 17 × 9 mm balloting masses were additionally visible, attached to its wall from the side of the pulmonary trunk (**A**,**B**). Computed tomography was performed, which describes the hypodense soft tissue formation adjacent to the upper wall of the left atrium, 14 × 16 mm with a wide base of 11 mm, and attached to the upper wall of the ear (**C**–**F**). The structure showed mobility consistent with that of the left atrium. Due to the morphological image of the formation (low density and irregular shape) and the possibility of a myxoma-like focal change, the patient was consulted by the Heart Team. Due to the ambiguous nature of the lesion (thrombus vs. myxoma), apixaban was changed to warfarin and qualified for MRI after 3 months. The examination was performed and there was no evidence of any mass in the left atrium, which confirmed that the found structure was a thrombus. The most common nodular changes in the heart are metastatic tumors of lung cancer, breast cancer and hematological neoplasms. Primary tumors are much rarer and are benign in 75% of cases (e.g., myxoma, fibroma, lipoma). Malignant neoplasms, mainly sarcomas and lymphomas, constitute 25% of primary tumors. Apart from proliferative changes in the heart, there are also non-neoplastic structures such as thrombus, vegetation or inflammatory tumors [1,2]. Echocardiography is the primary test used to diagnose additional masses in the heart due to its wide availability. Transthoracic echocardiography (TTE) allows for the assessment of the size, location and mobility of the mass. Transesophageal echocardiography (TEE) is especially valuable for assessing lesions located in the left atrium. Often, echocardiography enables the initial differentiation of myxoma from thrombus based on differences in tissue perfusion [3,4]. However, in some patients, more accurate imaging techniques, such as computed tomography or magnetic resonance imaging, are necessary in the differential diagnosis [5,6,7].

## Data Availability

Original data supporting the reported results are available contacting the authors.

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
