# Peer review of "Echocardiography, Computed Tomography and Magnetic Resonance Imaging in the Differential Diagnosis of a Tumor in the Left Atrium of the Heart"

_diagnostics, 2022, doi:10.3390/diagnostics12071749_

Round 1

Reviewer 1 Report

The differential diagnosis of myxoma vs thrombus in the atria is of extreme importance especially for the patients' follow-up. Moreover, cardiac tumors are very rare but the awareness of this possibility must always be present in the Heart Team and radiologists included.

This work by Zalewska-Adamiec et al. has this aim: to present to the clinical community the presence of these nosological entities and how to deal with them from a diagnostic point of view through the imaging techniques available to date.

Author Response

Thank you for your nice comment.

Best regards.

Reviewer 2 Report

Nice work

Author Response

(The authors gave the same response as above.)

Reviewer 3 Report

some language changes should be made.

Overall, the present manuscript is of interest fpr the special issue.

However, the authors should mentione in the introduction and discussion some intracardiac lesions:

primary and secondary heart lymphoma (for example see the following:

doi: 10.1177/0003319717713581);

primary heart sarcoma (for example: doi: 10.1111/jocs.15538)

primary cardiac lipoma (10.1186/s13019-020-01379-6)

other primary cardiac tumors ( doi: 10.1007/s00246-022-02814-2).

Author Response

Dear Reviewer

Thank you for your valuable comments.

Language changes have been made.

In the abstract and in the commentary to the Figures, fragments about other heart cancers were added.

Changes in the text are marked in red

Best regards.

Round 2

Reviewer 3 Report

none